

# Quenching the Kitaev honeycomb model

Louk Rademaker[1,2⋆]

**1** Department of Theoretical Physics, University of Geneva, 1211 Geneva, Switzerland
**2** Perimeter Institute for Theoretical Physics, Waterloo, Ontario N2L 2Y5, Canada

⋆ louk.rademaker@gmail.com

## Abstract

I studied the non-equilibrium response of an initial Néel state under time evolution with the Kitaev honeycomb model. With isotropic interactions ($J_x = J_y = J_z$) the system quickly loses its antiferromagnetic order and crosses over into a steady state valence bond solid, which can be inferred from the long-range dimer correlations. There is no signature of a dynamical phase transition. Upon including anisotropy ($J_x = J_y \neq J_z$), an exponentially long prethermal regime appears with persistent magnetization oscillations whose period derives from an effective toric code.

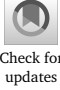
# 1 Introduction

Quantum spin liquids [1] are intriguing forms of matter characterized by the absence of magnetic order and the presence of long-range entanglement. A defining feature is that they cannot be transformed smoothly into a non-entangled magnetic product state, such as the Néel antiferromagnet. One might wonder whether these opposite extremes can be connected under a *rapid* change of external parameters.

Such a rapid change is known as a quench [2,3], and this set-up has lead to the prediction and observation of dynamical phase transitions. [4–6] For example, in the transverse field Ising model, time evolution of an initial magnetic state under a Hamiltonian with a trivial paramagnetic ground state leads to nonanalytic behavior in the return amplitude at certain times after the quench. [4] Also the opposite quench, starting from a paramagnetic spin liquid and time-evolving with a Hamiltonian not supporting spin liquid behavior, has been studied [7]. In this work, I will combine these works to answer the question: what happens when time evolves a magnetic state with a Hamiltonian that has a spin liquid ground state? Will we see a dynamical phase transition or crossover into a spin liquid regime at some finite timescale? Or will signatures of the initial magnetic order remain?

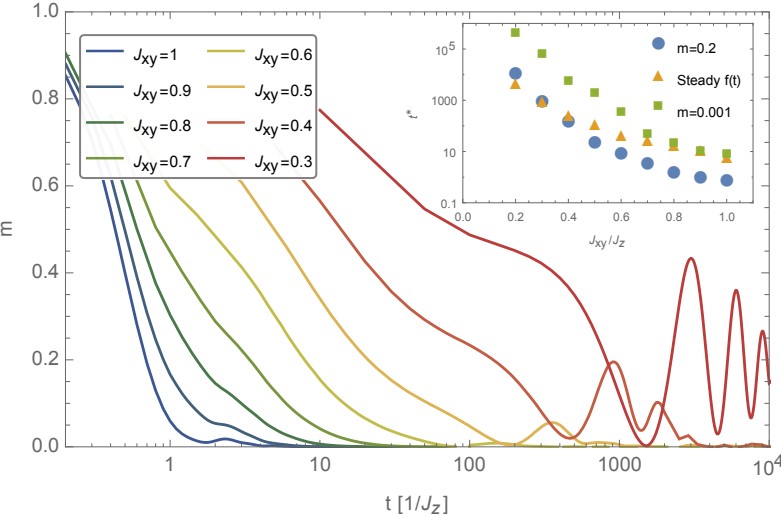

Figure 1: The staggered magnetization $m = (-1)^i \langle \sigma_i^z(t) \rangle$ after the quench for various $J_{xy}$, fixed $J_z = 1$, $N_{mc} = 2000$, and system size $L = 8$. While the magnetization vanishes quickly in the isotropic model, the response is exponentially slower when $J_{xy} < 1$. For completeness, the system size dependence of the staggered magnetization for $J_{xy} = 0.2 J_z$ is shown in Fig. 4. *Inset:* Typical timescales as a function of anisotropy. Shown here are the times it takes for the system to lose 80% and 99.9% of its staggered magnetization, as well as the time where the free energy density reaches its steady state value.

The Kitaev honeycomb model [8] provides an ideal playground to answer this question since it is exactly solvable. A slow ramp in this model has been studied before [9,10], but there the dynamics started from an initial spin liquid state. Here, I start from an antiferromagnetic Néel state, the simplest possible non-entangled magnetically ordered state, and time evolve with the Kitaev Hamiltonian with both isotropic ($J_x = J_y = J_z$) and anisotropic interactions ($J_x = J_y \neq J_z$). In order to time evolve with the Kitaev model I first express the Néel state as a superposition of different gauge fields configurations. Within each gauge sector, I then compute the exact time evolution of the free Majorana fermions.

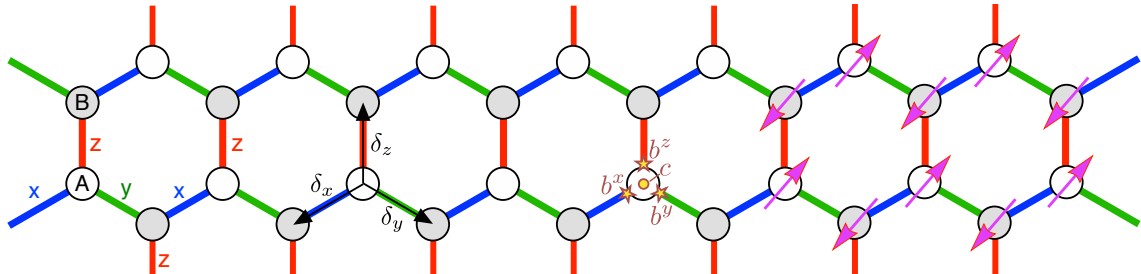

Figure 2: The Kitaev honeycomb model on a lattice. The unit cell with sites $A$ and $B$ is shown, together with the three inequivalent bonds labeled $\alpha = x, y, z$. The vectors $\delta_\alpha$ indicate the nearest neighbor position relative to an A site. In the middle of the lattice I indicate how a spin can be split up into four Majorana operators $b^\alpha$ and $c$. On the right a visualization of the initial Néel state.

As expected, the initial staggered magnetization vanishes (see Fig. 1) after the quench. Unlike quenches in the transverse field Ising model, however, there seem to be no signatures of a dynamical phase transition. At long times the system becomes a steady state valence bond solid. More surprisingly, an exponentially long prethermal regime appears when the interactions are anisotropic, as seen for example in the time evolution of the magnetization (Fig. 1). This prethermal regime is governed by an effective high-temperature toric code.

In Sec. 2 I will first present the Kitaev honeycomb model, its exact solution and an outline of the quench method. The results are discussed in Sec. 3, with a special emphasis on the question of a dynamical phase transition (3.1), the prethermal regime (3.2) and the final steady state (3.3). I conclude with a brief discussion on entanglement and experimental realizations in Sec. 4.

## 2 Model, initial state and method

Before presenting the results in detail, let me introduce the set-up of the quench. Consider spin-$\frac{1}{2}$ degrees of freedom $\sigma_i$ on a honeycomb lattice. The unit cell has two sites, which I will label as the $A$ and $B$ site, shown in Fig. 2. The initial state will be a perfect Néel state polarized along the $z$-direction, which is an unentangled product state $|\psi_0\rangle = \prod_i |\uparrow_{iA}\rangle \otimes |\downarrow_{iB}\rangle$. Starting from this initial state I will compute the time evolution using the Kitaev honeycomb model. In this model the bonds between lattice sites are divided into three types, depending on their direction, as shown in Fig. 2. Each bond-type has an Ising spin interaction along a different spin orientation,

$$H = \sum_i \left( J_x \sigma^x_{iA} \sigma^x_{i+\delta_x, B} + J_y \sigma^y_{iA} \sigma^y_{i+\delta_y, B} + J_z \sigma^z_{iA} \sigma^z_{iB} \right). \tag{1}$$

Kitaev's key insight was that one can solve this model exactly by representing each spin by four Majorana operators $b^x, b^y, b^z$ and $c$. This enlarges the Hilbert space, and in the enlarged Hilbert space we can define 'enlarged' spin operators $\tilde{\sigma}^x = ib^x c, \tilde{\sigma}^y = ib^y c$, and $\tilde{\sigma}^z = ib^z c$. The projection operator onto the real, physical, subspace is $P = \frac{1}{2}(1 + b^x b^y b^z c)$. Therefore, the physical spins are given by $\sigma^\alpha = P\tilde{\sigma}^\alpha P$, which implies $\sigma^x = \frac{i}{2}(b^x c - b^y b^z), \sigma^y = \frac{i}{2}(b^y c - b^z b^x)$, and $\sigma^z = \frac{i}{2}(b^z c - b^x b^y)$. In the following, I will use that within the physical subspace, the real spins can also be represented by the operators of the form $\sigma^z = -ib^x b^y$ and similar expressions hold for $\sigma^x$ and $\sigma^y$.

In terms of the new Majorana operators, the Hamiltonian reads

$$H = i \sum_{j,\alpha} J_\alpha u_{j\alpha} c_{jA} c_{j+\delta_\alpha,B} \, , \tag{2}$$

where $j$ sums over unit cells and $u_{j\alpha} = i b_{jA}^\alpha b_{j+\delta_\alpha,B}^\alpha = \pm 1$ is a static $Z_2$ gauge field living on the $\alpha = x, y, z$ bond. The product of $Z_2$ gauge fields along a plaquette is gauge-invariant and is the 'flux' $w_p = \sigma_1^x \sigma_2^y \sigma_3^z \sigma_4^x \sigma_5^y \sigma_6^z$. The remaining $c$-Majorana's are called 'matter' and are noninteracting.

The spin liquid ground state of the Kitaev honeycomb model is in the zero-flux sector, meaning all gauge fields $u_{j\alpha}$ are the same. In contrast, the Néel state, when expressed in terms of gauge and matter fields, is in a superposition of different flux configurations since $\langle \psi_0 | w_p | \psi_0 \rangle = 0$ where $w_p$ is the plaquette flux operator. We can show which flux configurations are included in this superposition by repeated use of the fact that the Néel state is an eigenstate of the physical operator $\sigma_i^z$.

A good basis to describe the Néel state is by pairing the remaining matter Majorana's along the $z$-bonds within one unit cell, $v_j = i c_{jA} c_{jB} = \pm 1$. Any possible state in the enlarged Hilbert space can be written as a superposition of $u, v$-configurations, $|\psi\rangle = \sum_{\{u_{j\alpha}, v_j\}} c_{\{u_{j\alpha}, v_j\}} | \{u_{j\alpha}, v_j\} \rangle$, and our task is to find the weight constants $c_{\{u_{j\alpha}, v_j\}}$. The fact that the Neel state is physical and therefore must satisfy $P_j |\psi_0\rangle = |\psi_0\rangle$, and that it is an eigenstate of $\sigma_j^z$ for every $j$, leads to two constraints on the possible $u, v$-configurations. On a lattice consisting of $L_x \times L_y$ unit cells with periodic boundary conditions, we have periodic chains of $xy$-bonds. The product of all $2L_x$ $z$-spins along such a $xy$-chain equals $(-1)$ times the product of all $x$ and $y$ gauge fields. Therefore, this product of gauge fields must equal $(-1)^{L_x+1}$. Consequently, the Néel state is an equal-weight superposition of all $N_c = 2^{3L_x L_y - L_y}$ possible $u_{j\alpha}$ gauge field configurations that satisfy this constraint. The matter content $v_i$ is fixed by the constraint $\sigma_{jA}^z \sigma_{jB}^z = -1$ within each unit cell, which implies $u_{jz} = v_j$. The relative phases between different $\{u_{j\alpha}, v_j\}$-configurations are fixed by the expectation value of $\sigma_j^z$ operators, and are multiples of $i$.

Note that in principle the gauge freedom allows us to construct the same Néel state with a different set of gauge field configurations. However, the current choice is extremely transparent since it represents the Néel state as an equal superposition of all allowed configurations. This in turn makes the calculation of observables straightforward.

Because the gauge fields are integrals of motion only the matter fields will be changing over time,

$$|\psi(t)\rangle = \frac{1}{\sqrt{N_c}} \sum_{\{u_{j\alpha}\}} |\{u_{j\alpha}\}\rangle \otimes e^{-iH^{\{u_{j\alpha}\}}t} |\psi_0^{\{u_{j\alpha}\}}\rangle \, , \tag{3}$$

where $\{u_{j\alpha}\}$ represents a gauge field configuration that respects the aforementioned constraints, $|\psi_0^{\{u_{j\alpha}\}}\rangle$ is the initial matter field configuration determined by $v_j = u_{jz}$ and $H^{\{u_{j\alpha}\}}$ is a free matter Majorana Hamiltonian with hoppings depending on the $Z_2$ gauge fields. The magnetization on an $A$ lattice site $m_{jA}(t) = \langle \psi(t) | \sigma_{jA}^z | \psi(t) \rangle$ can be found using the gauge-field-only representation of spin, $\sigma_{jA}^z = -i b_j^x b_j^y$. Therefore, the magnetization can be written as the return amplitude with *two* matter Hamiltonians,

$$m(t) = \frac{1}{N_c} \sum_{\{u_{j\alpha}\}} \langle \psi_0^{\{u_{j\alpha}\}} | e^{iH^{\{u'_{j\alpha}\}}t} e^{-iH^{\{u_{j\alpha}\}}t} | \psi_0^{\{u_{j\alpha}\}} \rangle \, , \tag{4}$$

where the configurations $\{u'_{j\alpha}\}$ and $\{u_{j\alpha}\}$ differ only by the flip of the two gauge fields $u_j^x$ and $u_j^y$. The sum over exponentially many gauge field configurations can be replaced by a random Monte Carlo sampling over all configurations [11, 12] that satisfy the constraints relevant for

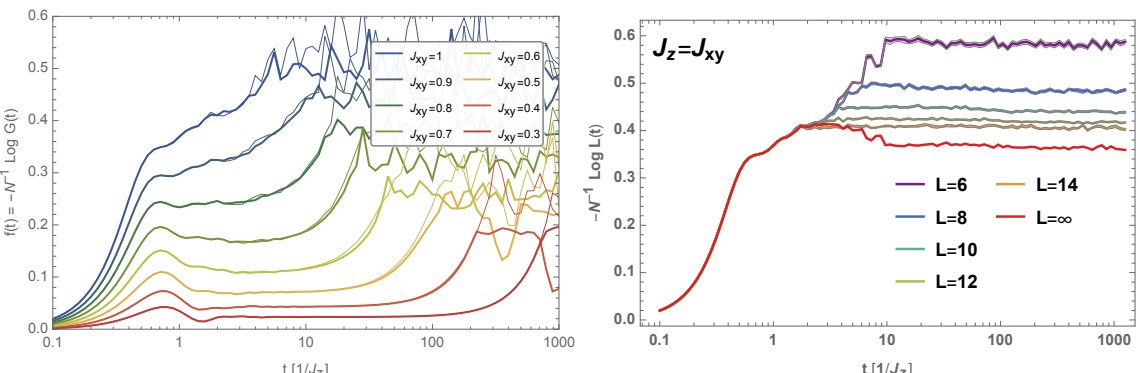

Figure 3: **Left:** The nonequilibrium free energy $f(t) = -\frac{1}{N}\log\mathcal{G}(t)$ for various $J_{xy}$, fixed $J_z = 1$, $N_{mc} = 20000$, and system sizes $L = 6$ (thin lines) and $L = 8$ (thick lines), where $L_x = L_y = L$. No dynamical phase transition is observed in the short times before a steady state plateau emerges. A prethermalization regime appears with increasing anisotropy. **Right:** The nonequilibrium free energy for the isotropic case $J_z = J_{xy}$ as a function of system size $L$, averaged over $N_{mc} = 500,000$ gauge configurations. I extrapolated these results to $L = \infty$, which still does not show any signatures of nonanalytic behavior.

the initial Néel state. For each such configuration I need to compute these generalized return amplitudes for the matter Hamiltonian, which can be done efficiently using the Balian-Brezin decomposition as outlined in Appendix A. [13, 14] Note that an alternative way of deriving my results is by using the 'brick wall'-representation of the Kitaev honeycomb model. [15]

Note that in the basis we use, where we pair Majorana matter particles along the $z$-bonds to create complex fermions, it is most natural to compute expectation values of $S^z$ and correlations thereof. In contrast, the computation of correlations functions containing $S^x$ or $S^y$ is more involved. Specifically, such correlation functions can no longer be expressed purely in terms of gauge fields $b$, and would require keeping track of the time evolution of Majorana matter fermions. Since we start with a $z$-polarized Néel, we only focus in this manuscript on the dynamics of correlation functions involving only $S^z$ operators.

Finally, it is worth mentioning that the sampling over all gauge fields is specific to our choice of initial state. In the extreme case that one has an initial state that lies purely within one flux sector, the corresponding quench dynamics is that of a non-interacting fermion model. This is done in, for example, Refs. [9, 10], where they study a ramp within the zero-flux sector. In this case the dynamics follow the general lore of dynamical phase transitions [4–6, 16]. Much of our results in the following section deviate from this precisely because we intertwined various flux sectors by choosing an initial Néel state.

## 3 Results

I will now show results for a quench from an initial Néel antiferromagnet, to the Kitaev honeycomb model. I consider both quenches to the isotropic case ($J_x = J_y = J_z$) as well as the anisotropic Kitaev model ($J_x = J_y \neq J_z$). The anisotropy is defined by the ratio $J_z/J_{xy}$ where $J_{xy} \equiv J_x = J_y$.

### 3.1 Phase transition or crossover?

The dynamics studied here can be viewed as a quench through a quantum critical point separating an antiferromagnetic phase and a paramagnetic spin liquid phase. It has been suggested that a quench from the ferromagnetic to the paramagnetic phase leads to non-analytic behavior of the return amplitude at given times. [4–6, 16] Specifically, consider the nonequilibrium free energy density

$$f(t) = -\frac{1}{N} \log |\mathcal{G}(t)| \,, \tag{5}$$

where $N = L_x L_y = L^2$, and $\mathcal{G}(t)$ is the return amplitude or *Loschmidt echo*

$$\mathcal{G}(t) = \langle \psi(t) | \psi_0 \rangle. \tag{6}$$

In the transverse field Ising model, this quantity is shown to be nonanalytical at several moments after the quench. Such nonanalytic points are called *dynamical phase transitions*, in analogy to thermal phase transitions where the free energy becomes nonanalytic.

In order to study the possible appearance of a dynamical phase transition in our quench model, I computed the nonequilibrium free energy density $f(t)$. The results are shown in Fig. 3. On the left-hand side I show the free energy, for $J_z = 1$ and various $J_{xy}$, averaged over $N_{mc} = 20000$ gauge configurations. Independent of $J_{xy}$, there is an initial growth of free energy. Subsequently, a plateau (discussed in Sec. 3.2) appears for the anisotropic cases. After that, there seem to be severe system-size fluctuations and it is not apparent whether or not a true nonanalyticity appears.

On the right of Fig. 3 I show the system-size dependence of the free energy for the isotropic case, computed using much more gauge configurations ($N_{mc} = 500,000$). Around the time $t = 10$ there seems to be a steady state plateau developing for the free energy, which is strongly system size dependent. The infinite $L$ limit, however, does not seem to suggest any nonanalytic behavior. The evolution of the free energy is likely more accurately described as a crossover.

It is important to note that many dynamical phase transitions have been found in models that are noninteracting, such as the transverse field Ising model, [4–6, 16] the XY model, [17] or fermionic band insulators. [18] Even though the Kitaev model is exactly solvable, it is not a noninteracting theory. This might be the reason why I do not observe any dynamical phase transition.

### 3.2 Prethermalization

Upon increasing the anisotropy $J_z/J_{xy}$, a plateau emerges in the free energy (Fig. 3) that lasts long in the anisotropy ratio $J_z/J_{xy}$. During this exponentially long timescale, persistent oscillations in the staggered magnetization $m(t) = \sum_j (-1)^j \langle \sigma_j^z(t) \rangle$ appear. To emphasize this behavior, we show in Fig. 4 the staggered magnetization for $J_{xy} = 0.2 J_z$ as a function of system size, including a $L = \infty$ limit. Even though the anisotropy is only $J_z/J_{xy} = 5$, the time-scale over which the magnetization persists is about $10^5$ longer than for the isotropic case. Different measures of a typical time-scale, namely the onset of the free energy plateau or when the magnetization reaches a 0.2 or 0.001 threshold, all display an approximately exponential dependence on the anisotropy $t^* \sim e^{cJ_z/J_{xy}}$, as is shown in the inset of Fig. 1.

Both of these phenomena - the long time-window $t^*$ and the persistent magnetization oscillations - can be understood within the framework of *prethermalization*. [20–25]

Let us first consider the length of the time-scale $t^*$. In typical quenched systems, dynamical time-scales would depend in a power-law fashion on an anisotropy parameter. For example, in a quench of the transverse field Ising model from the ferromagnetic to the paramagnetic phase, the typical time-scale is set by $t^* = \pi/\epsilon_{k^*}(g_1)$, where $\epsilon_k(g) = \sqrt{(g - \cos k)^2 + \sin^2 k}$,

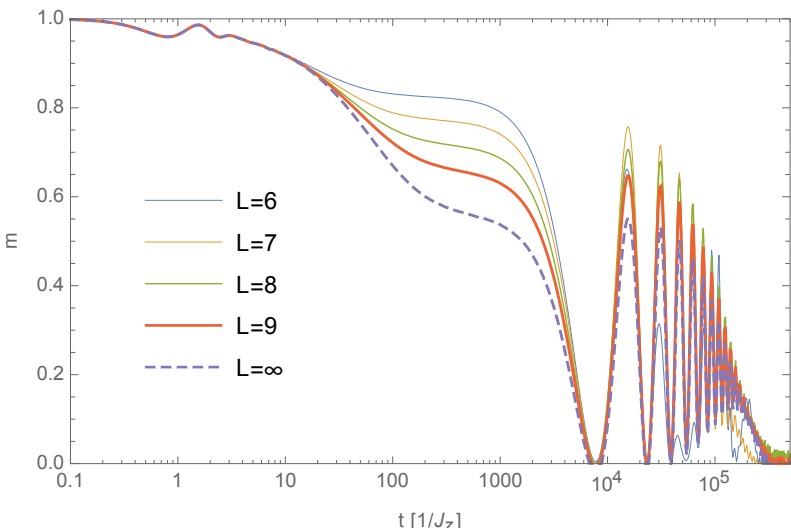

Figure 4: In case of large anisotropy the decay of magnetization is extremely slow, here shown for $J_{xy} = 0.2J_z$ with $N_{mc} = 2000$ and various system sizes extrapolated to $L = \infty$. Between $t_1 \sim 1$ and $t_2 \sim 10^{3.5}J_z^{-1}$ there is a persisting magnetization, due to the high return amplitude visible in Fig. 3. After this the system is dominated by large magnetization oscillations that finally disappear around $t_3 \sim 10^{5.5}J_z^{-1}$.

$\cos k^* = \frac{1+g_0 g_1}{g_0+g_1}$ and $g_0, g_1$ are the values of the transverse field before and after the quench. [4] This timescale diverges as $g_1$, the post-quench transverse field, becomes close to the critical value $g_c = 1$. It is easy to show that this divergence indeed follows a power-law, $t^* \sim (g_1 - g_c)^{-1/2}$.

So why is the time-scale $t^*$ so much longer in the case of quenching the anisotropic Kitaev model? The answer lies in the concept of prethermalization [20–25] that occurs in systems close to integrability. In particular, the dynamics here can be understood using the framework of Ref. [23]. For the anisotropic Kitaev model, the coupling along the $z$-bonds is significantly stronger than along the $x, y$-bonds. We can therefore treat the $x, y$-bond coupling as a perturbation. The Kitaev model is written as

$$\hat{H} = -J_z \hat{N} + J_{xy} \hat{Y} \, , \tag{7}$$

where $\hat{N}$ is the sum of all the $z$-bond couplings, and $\hat{Y}$ contains the couplings along the $x, y$-bonds. The term $\hat{N}$ is trivially integrable, it is just a sum of local commuting terms with integer eigenvalues. Following Ref. [23], for $J_{xy} < J_z$, we can perform a unitary transformation such that the Hamiltonian becomes

$$\hat{H} = -J_z \hat{N} + \hat{H}'_{\text{eff}} + \mathcal{O}(e^{-J_z/J_{xy}}) \, , \tag{8}$$

where the new term $\hat{H}'_{\text{eff}}$ commutes with $\hat{N}$. This means that $\hat{H}'_{\text{eff}}$ does not affect the relative orientation of two spins along a $z$-bonds. More importantly, the remaining term is exponentially small, meaning that for an *exponentially long time* the dynamics preserve the spin correlations along each $z$-bond. To summarize, an exponentially long timescale $t^* \sim e^{cJ_z/J_{xy}}$ appears because for a suitable unitary transformation the Hamiltonian has effectively *new conservation laws* that constrain dynamics for a long time.

The fact that the spin correlations are locked along a $z$-bond can be inferred from measuring the static spin correlation function $S_{ij}^{zz}(t) = \langle \psi(t)|\sigma_i^z \sigma_j^z|\psi(t)\rangle$. As shown in Fig. 5, even in the isotropic case the relative orientation of spins along a $z$-bond remains nonzero in

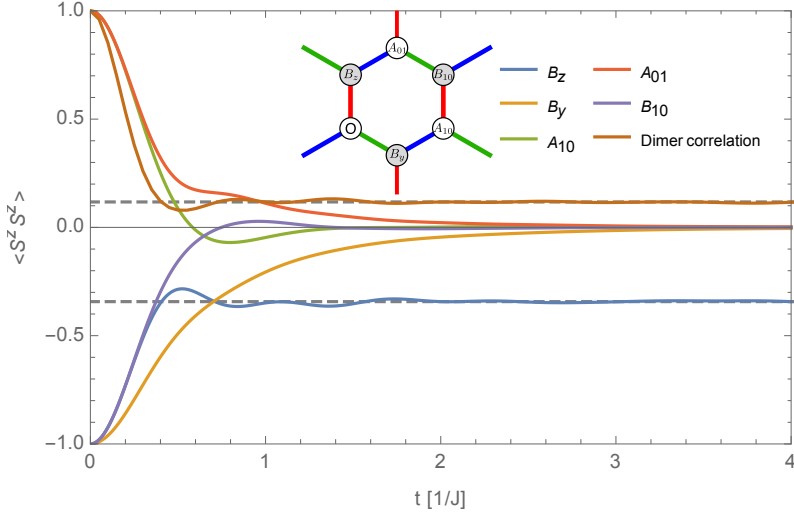

Figure 5: The static spin-spin correlations $\langle \sigma_i^z \sigma_j^z \rangle$ for various short-range spins, in the isotropic model, with $L = 8$ and $N_{mc} = 2000$. After a short time all spin correlations vanish except the nearest-neighbor correlation along a $z$-bond. The long-range dimer-dimer correlation function, here measured at the longest possible distance between two sets of $z$-bonds, also obtains a nonzero steady state value. These correlations are indicative of a valence bond solid phase. [19]

the infinite time-limit. This can be further corroborated by computing the static spin correlations in the diagonal ensemble (see Appendix B), which indeed yields a steady state with zero magnetization but nonzero spin correlations along the $z$-bond. Notice that other static spin correlations vanish.

Having established the $z$-bond spin-lock for an exponentially long time, we can investigate how this leads to persistent magnetization oscillations during this prethermal regime. The key lays in the effective Hamiltonian $\hat{H}'_{\text{eff}}$ in Eq. (8), which describes the effective interaction between the locked spins along a $z$-bond. At each $z$-bond, the configuration must be antiferromagnetic, meaning that only the spin configurations ↑↓ and ↓↑ are allowed. These two states constitute a 'new spin' $\tau$, and using Kitaevs fourth order perturbation theory [8] the effective Hamiltonian becomes

$$\hat{H}'_{\text{eff}} = -\frac{J_{xy}^4}{16 J_z^3} \sum_p \tau_{p,\text{left}}^y \tau_{p,\text{top}}^z \tau_{p,\text{right}}^y \tau_{p,\text{bottom}}^z + \cdots , \tag{9}$$

where $p$ is every plaquette of the honeycomb lattice, and left/right/etc. refers to the $z$-bond to the left/right/etc. of this plaquette. Note that this model is equivalent to the toric code. [26] Any attempts to understand the dynamics in terms of topology are futile, as we are very far from the ground state during our quench dynamics and signatures of topology, such as anyonic excitations, are defined only close to the ground state.

Nonetheless, a simple calculation shows what would happen if one starts with an initial Néel state (meaning ↑↓ on every $z$-bond) and let time evolve with the Hamiltonian of Eq. (9). In such a quench, the staggered magnetization will oscillate according to $m(t) = \cos^2(2J_{\text{eff}}t)$ where $J_{\text{eff}} = \frac{J_{xy}^4}{16 J_z^3}$. Similarly, ignoring higher-order corrections to Eq. (9), in the prethermal regime the effective toric code will cause *persistent oscillations* with period

$$T = \frac{8\pi J_z^3}{J_{xy}^4} . \tag{10}$$

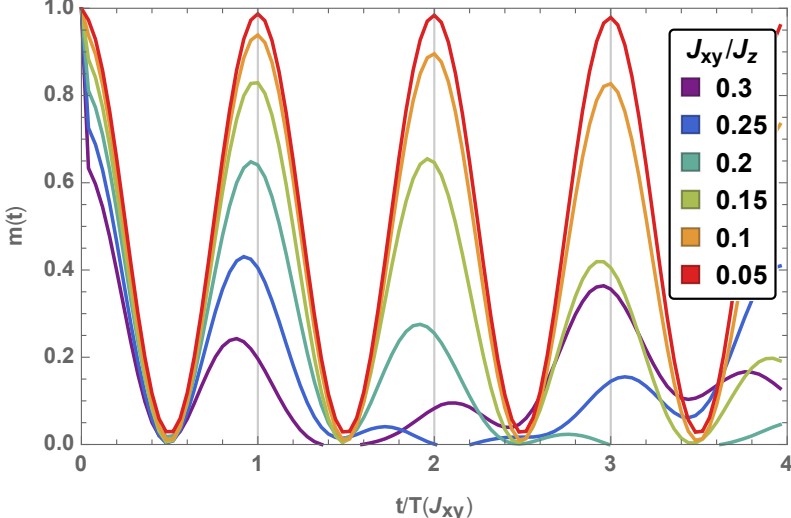

Figure 6: In the anisotropic limit $J_{xy} \ll J_z$, the exponentially long prethermal regime is governed by an effective toric code. This causes persistent oscillations in the magnetization with a period $T(J_{xy})$ given by Eq. (10). Here we show the magnetization oscillations in the range $J_{xy}/J_z = 0.05 - 0.3$ for linear system size $L = 6$, with $N_{mc} = 2000$. The time (horizontal axis) is rescaled for each value of $J_{xy}$ to correspond to exactly four oscillation periods. We find that indeed for small $J_{xy}$, we approach the predicted periodicity.

The theory of prethermalization thus predicts that in quenching the anisotropic Kitaev model, there is a regime that for an exponentially long timescale $t^* \sim e^{J_z/J_{xy}}$ during which you will see persistent magnetization oscillations between nonzero $m$ and 0 of period $T \sim \frac{J_z^3}{J_{xy}^4}$.

To confirm this prediction, we analyzed how the period of persistent oscillations varies with $J_{xy}/J_z$. The results are shown in Fig. 6 for $L = 6$, $N_{mc} = 2000$ and varying small $J_{xy}$. Whenever $J_{xy} < 0.2J_z$, the magnetic oscillations have a period that is well approximated by Eqn. (10). We conclude that indeed, for an exponentially long time, the system undergoes magnetic oscillations as dictated by the toric code.

## 3.3 Steady state valence bond solid

In the isotropic case there is no signature of prethermalization and after a short time of order unity the system equilibrates. While there is zero net staggered magnetization in this steady state, there are remnant nearest-neighbor spin correlations along the $z$-bond, as shown in Fig. 5. This suggests the steady state is a valence bond solid with the singlets oriented along the z-bonds. To further corroborate this claim, I studied the dimer-dimer correlation function $D_{ij}^{zz}(t) = \langle \sigma_i^z \sigma_{i+\delta_z}^z \sigma_j^z \sigma_{j+\delta_z}^z \rangle$ that has been used before as an indication of valence bond order. [19] Indeed, I find long-range dimer order, even though the state is at relatively high temperatures. Notice that this state does break rotational invariance, since the singlet bonds live on the $z$-bonds which are inequivalent to the $x, y$-bonds of the lattice.

Another way to quantify the steady state is through the dynamic two-time spin correlation function $S_j^{zz}(t, t') = \langle \psi(t) | \sigma_{jA}^z(t') \sigma_{jB}^z | \psi(t) \rangle$. [27–29] The Fourier transform with respect to $t' - t$ can be interpreted as an AC spin conductivity. Specifically, the DC ($\omega = 0$) response measures the antiferromagnetic correlations along a $z$-bond. For small $\omega$ the correlations are reduced over a frequency scale set by the flux-averaged Majorana density of states, see Fig. 7. At later times this correlations gets suppressed in the frequency range between 0 and

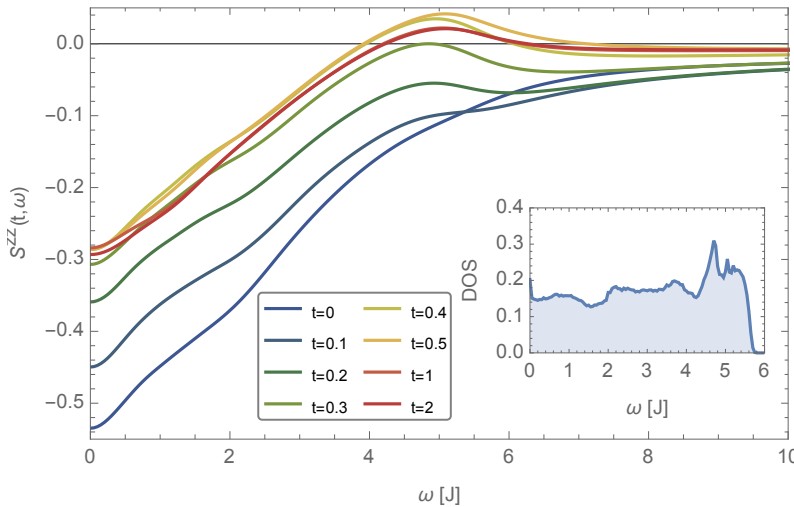

Figure 7: The dynamical two-time response in the isotropic quench, as a function of frequency $\omega$ for various waiting times $t$. Directly after the quench ($t = 0$) the peak at $\omega = 0$ indicates strong antiferromagnetic correlations. This peak is suppressed as time progresses, consistent with the loss of magnetization, see Fig. 1. After $t \sim 0.3 J^{-1}$ the small frequency peak remains constant, and a dynamic magnetization reversal occurs in the frequency range between $4 - 6$ J, where the flux-averaged Majorana density of states (see inset) is highest and corresponding to the triplet excitation in the valence-bond solid.

6J, which is the flux-averaged bandwidth of the matter Majorana's. Interestingly, for times $t > 0.5$ a reversal of the dynamic correlations for $4J < \omega < 6J$ appears. This is a signature of the elementary triplet excitation of the valence bond solid.

We thus find a dynamic crossover from a Néel state to a valence bond solid. This transition in equilibrium is known as a deconfined quantum phase transition and falls outside the usual Landau classification of continuous phase transitions. [30] The absence of any finite time singularity is due to the fact that the location of the valence bonds are determined by the orientation of the initial Néel state. There is no dynamical spontaneous symmetry breaking: a Néel state polarized along the $x$-axis would give rise to a valence bond solid with singlets along the $x$-bonds, and so forth. It is an interesting open question to study what happens when an initial Néel state is not aligned along one of the principal spin axes.

## 4 Conclusion and Discussion

I showed that starting from a Néel state, time evolution with the Kitaev honeycomb model leads to crossover to a steady state valence bond solid. When the interactions are anisotropic ($J_z/J_{xy} \neq 1$), an exponentially long prethermal regime appears whose dynamics can be effectively described by a toric code.

Note that similar results are expected if one would start with an initial ferromagnetic product state, rather than an antiferromagnetic product state.

To what extent these results remain valid beyond the exactly solvable model, for example by introducing a small Heisenberg term, is an open question. Based on the proof of Ref. [23] I expect that the prethermal regime will persist even in the presence of such perturbations, but quantifying this requires new computational techniques beyond the ones used in this work.

An interesting aspect that was not included in this study is the topological nature of the

Kitaev honeycomb model. The ground state of the model has nontrivial entanglement entropy, [31,32] a topological groundstate degeneracy and has anyonic excitations. [8] It is hard, however, to see one of these topological signatures in the quench set-up. After all, the Néel state has a high energy density expectation value in the Kitaev honeycomb model, meaning that the quench dynamics are far away from the ground state. The generated entanglement is therefore volume law, making it almost impossible to detect a possible topological entanglement entropy. The same holds for anyonic excitations, which are well-defined only close to the ground state. Interestingly, topological degeneracy might be observed. On a torus, the Kitaev honeycomb model in the toric code regime has a fourfold degeneracy, meaning that there are orthogonal ground states that are locally indistinguishable. Acting with a string of spin operators allows you to go from one to the other ground state. Now in the final valence bond steady state, acting with a vertical string of $\sigma^x$ and $\sigma^y$ operators will create an orthogonal state that is indistinguishable from the original state on the level of single-spin operator measurements. I leave it for future work to investigate whether indeed this amounts to a topological degeneracy, which might be relevant for quantum computations.

Finally, in recent years some material systems have been proposed to be experimental realizations of the Kitaev honeycomb model [33]. Though straining these materials is unlikely to give rise to the desired anisotropy to observe a prethermal regime, it might be possible to chemically engineer these system to get the anisotropic interactions desired. It will also be interesting to see the dynamic response after a quench with an initial state resembling the spiral magnetic order found in these materials. [34,35]

# Acknowledgements

We thank Tim Hsieh, Khadijeh (Sona) Najafi, Leon Balents, Hae-Young Kee, Yong Baek Kim and Zohar Nussinov for useful discussions.

**Funding information** This research was supported by Perimeter Institute for Theoretical Physics. Research at Perimeter Institute is supported by the Government of Canada through the Department of Innovation, Science and Economic Development Canada and by the Province of Ontario through the Ministry of Research, Innovation and Science.

# A    Matter Hamiltonian time evolution

As described in the main text, the time evolution of the Kitaev honeycomb model is completely due to the Majorana fermions. In each unit cell $j$, which contains a $z$-link, we indentify an $A$ and $B$ sublattice site. The $c$-Majorana's in Kitaev's notation are then paired along the $z$-link to form complex fermions,

$$c_{jA} = a_j + a_j^\dagger, \tag{11}$$

$$c_{jB} = -i(a_j - a_j^\dagger). \tag{12}$$

The matter Hamiltonian on the full honeycomb lattice reads

$$
\begin{aligned}
H^{\{u\}} = & -\sum_j \Big\{ (J_z u_j^z)(2a_j^\dagger a_j - 1) \\
& + (J_x u_j^x)(a_j + a_j^\dagger)(a_{j+\delta_x} - a_{j+\delta_x}^\dagger) + (J_y u_j^y)(a_j + a_j^\dagger)(a_{j+\delta_y} - a_{j+\delta_y}^\dagger) \Big\},
\end{aligned} \tag{13}
$$

where $j$ labels a unit cell, and $\delta$ connects to the unit cell with center at position $\delta_x = \frac{1}{2}(-\sqrt{3}\hat{x} - 3\hat{y})$, and $\delta_y = \frac{1}{2}(\sqrt{3}\hat{x} - 3\hat{y}))$.

In each gauge sector, the required initial state is the product state where unit cells with $u_j^z = 1$ are occupied with a complex matter fermion. For later purposes it is practical to perform a particle-hole transformation on 'occupied' sites, so that the Hamiltonian becomes

$$H^{\{u\}} = \sum_j \Big\{ J_z(2a_j^\dagger a_j - 1) + (J_x u_{j+\delta_x}^z u_j^x)(a_j + a_j^\dagger)(a_{j+\delta_x} - a_{j+\delta_x}^\dagger) \\ + (J_y u_{j+\delta_y}^z u_j^y)(a_j + a_j^\dagger)(a_{j+\delta_y} - a_{j+\delta_y}^\dagger) \Big\}. \tag{14}$$

With the Hamiltonian Eqn. (14), the initial matter state is nothing but the $a$-vacuum $|0\rangle$, defined by $a_j|0\rangle = 0$. This matter Hamiltonian can be brought into a canonical Bogoliubov-De Gennes (BdG) format,

$$H^{\{u_{j\alpha}\}} = \frac{1}{2} \begin{pmatrix} \mathbf{a}^\dagger & \mathbf{a} \end{pmatrix} \begin{pmatrix} H_d & \Delta \\ -\Delta & -H_d \end{pmatrix} \begin{pmatrix} \mathbf{a} \\ \mathbf{a}^\dagger \end{pmatrix}, \tag{15}$$

where $H_d$ is a real-valued symmetric matrix, $\Delta$ a real-valued antisymmetric matrix, and the vector $\begin{pmatrix} \mathbf{a}^\dagger & \mathbf{a} \end{pmatrix}$ contains all creation and annihilation operators for all unit cells. The $2N \times 2N$ BdG matrix in Eqn. (15) can be diagonalized, $H_{BdG} = V\Lambda V^\mathsf{T}$, with real eigenvalues $\Lambda = \mathrm{diag}(\epsilon_1, \epsilon_2, \dots, \epsilon_N, -\epsilon_1, \dots, -\epsilon_N)$ and $V$ a real orthogonal matrix of the form $V = \begin{pmatrix} Q & R \\ R & Q \end{pmatrix}$.

This diagonalization allows us to compute the Balian-Brezin decomposition of the time evolution operator, [13, 14]

$$e^{-iHt} = e^{\frac{1}{2}a^\dagger X a^\dagger} e^{a^\dagger Y a} e^{\frac{1}{2}a Z a} \det\big[Re^{-i\Lambda t/2} + Qe^{i\Lambda t/2}\big], \tag{16}$$

where $A = Qe^{i\Lambda t}Q^\mathsf{T} + Re^{-i\Lambda t}R^\mathsf{T}$, $B = Qe^{i\Lambda t}R^\mathsf{T} + Re^{-i\Lambda t}Q^\mathsf{T}$, $X = BA^{-1}$, $e^{-Y^\mathsf{T}} = A$, and $Z = A^{-1}B^*$.

The simplest quantity to compute is the overlap of the initial state with the time-evolved state, known as the return amplitude $\mathcal{G}(t) = \langle \psi(t)|e^{-iHt}|\psi_0\rangle$. Because different flux sectors are orthogonal to one another, the total return amplitude is a sum of matter Majorana return amplitudes in each gauge sector,

$$\mathcal{G}(t) = \frac{1}{N_c} \sum_{\{u_{j\alpha}\}} \langle \psi_0^{\{u_{j\alpha}\}}|e^{-iH^{\{u_{j\alpha}\}}t}|\psi_0^{\{u_{j\alpha}\}}\rangle. \tag{17}$$

Note that due to the particle-hole transformation, the state $|\psi_0^{\{u_{j\alpha}\}}\rangle$ is equal to the $a$-vacuum, so $a_j|\psi_0^{\{u_{j\alpha}\}}\rangle = 0$ for every $a_j$. To simplify notation, from now on I will write $|0\rangle$ for the initial state.

The return amplitude for a single free Majorana Hamiltonian follows directly from the Balian-Brezin decomposition Eqn. (16),

$$\langle 0|e^{-iH^{\{u_{j\alpha}\}}t}|0\rangle = \det\big[Re^{-i\Lambda t/2} + Qe^{i\Lambda t/2}\big]. \tag{18}$$

Since the number of gauge field configurations scales exponentially with system size, it is impossible to compute the above sum of Eqn. (17) exactly. Instead, I averaged over $N_{mc}$ random gauge field configurations that satisfy the constraints set by the initial state. It turns out that $N_{mc} = 1000$ yields sufficient accuracy for the system sizes considered.

The staggered magnetization, defined as $m(t) = \frac{1}{2N} \sum_j \langle \psi(t)|\sigma_{jA}^z - \sigma_{jB}^z|\psi(t)\rangle$, will decay over time starting from $m(t=0) = 1$. Using the representation $\sigma_{jA}^z = -ib_j^x b_j^y$, valid within

the physical subspace, we see that the magnetization can be computed as a sum over return amplitudes involving two Hamiltonians,

$$R_2(t) = \langle 0|e^{iH_2 t}e^{-iH_1 t}|0\rangle \,, \tag{19}$$

where $H_1$ and $H_2$ only differ through a flip of the $u_{jx}$ and $u_{jy}$ gauge fields neighboring the spin that we want to measure. I proceed by making the Balian-Brezin decomposition for both $H_1$ and $H_2$,

$$\mathcal{R}(t) = \det\left[R_2 e^{i\Lambda_2 t/2} + Q_2 e^{-i\Lambda_2 t/2}\right]\det\left[R_1 e^{-i\Lambda_1 t/2} + Q_1 e^{i\Lambda_1 t/2}\right]\langle 0|e^{\frac{1}{2}aZ_2^* a}e^{\frac{1}{2}a^\dagger X_1 a^\dagger}|0\rangle \,. \tag{20}$$

The remaining part can be brought again in the Balian-Brezin form,

$$\begin{aligned}
\langle 0|e^{\frac{1}{2}aZ_2^* a}e^{\frac{1}{2}a^\dagger X_1 a^\dagger}|0\rangle &= \sqrt{\det\left[Z_2^* X_1 + I\right]} \tag{21}\\
&\times \langle 0|e^{\frac{1}{2}a^\dagger X_1 (Z_2^* X_1 + I)^{-1} a^\dagger}e^{a^\dagger(-\log(Z_2^* X_1 + I))^\intercal a}e^{\frac{1}{2}a(Z_2^* X_1 + I)^{-1} Z_2^*}|0\rangle \\
&= \sqrt{\det\left[Z_2^* X_1 + I\right]} \,. \tag{22}
\end{aligned}$$

The square root can be avoided by observing that both $Z$ and $X$ are skew-symmetric, and thus using the Sylvesters determinant lemma we find

$$\sqrt{\det\left[Z_2^* X_1 + I\right]} = \text{Pf}\left[\begin{pmatrix} X_1 & -I \\ I & Z_2^* \end{pmatrix}\right] \,, \tag{23}$$

where Pf[..] refers to the Pfaffian of that matrix. In my numerical simulations, I use the software from Ref. [36] to compute the Pfaffians.

In conclusion, the Balian-Brezin decomposition yields for the return amplitude with two Hamiltonians

$$R_2(t) = \det\left[R_2 e^{i\Lambda_2 t/2} + Q_2 e^{-i\Lambda_2 t/2}\right]\det\left[R_1 e^{-i\Lambda_1 t/2} + Q_1 e^{i\Lambda_1 t/2}\right]\text{Pf}\left[\begin{pmatrix} X_1 & -I \\ I & Z_2^* \end{pmatrix}\right] \,. \tag{24}$$

Note that the static correlations $S_{ij}^{zz}(t) = \langle\psi(t)|\sigma_i^z\sigma_j^z|\psi(t)\rangle$ can be computed using the same formule, where now the gauge fields need to be flipped on the $x, y$-bonds adjacent to both sites $i$ and $j$.

Finally, I can compute the dynamic two-time correlation function

$$S_{ij}^{zz}(t, t') = \langle\psi(t)|\sigma_i^z(t')\sigma_j^z|\psi(t)\rangle \,. \tag{25}$$

This requires the computation of a return amplitude of time evolution with three different Hamiltonians. Using repeatedly the Balian-Brezin trick this can be expressed as

$$\begin{aligned}
R_3(t, t') &= \langle 0|e^{iH_3(t+t')}e^{-iH_2 t'}e^{-iH_1 t}|0\rangle \tag{26}\\
&= \det\left[R_3 e^{i\Lambda_3(t+t')/2} + Q_3 e^{-i\Lambda_3(t+t')/2}\right]\det\left[R_2 e^{-i\Lambda_2 t'/2} + Q_2 e^{i\Lambda_2 t'/2}\right]\\
&\quad \times \det\left[R_1 e^{-i\Lambda_1 t/2} + Q_1 e^{i\Lambda_1 t/2}\right]\\
&\quad \times \langle 0|e^{\frac{1}{2}aZ_3^* a}e^{\frac{1}{2}a^\dagger X_2 a^\dagger}e^{a^\dagger Y_2 a}e^{\frac{1}{2}aZ_2 a}e^{\frac{1}{2}a^\dagger X_1 a^\dagger}|0\rangle \tag{27}\\
&= \det\left[R_3 e^{i\Lambda_3(t+t')/2} + Q_3 e^{-i\Lambda_3(t+t')/2}\right]\left(\det\left[R_2 e^{-i\Lambda_2 t'/2} + Q_2 e^{i\Lambda_2 t'/2}\right]\right)^{-1}\\
&\quad \times \det\left[R_1 e^{-i\Lambda_1 t/2} + Q_1 e^{i\Lambda_1 t/2}\right]\\
&\quad \times \text{Pf}\left[\begin{pmatrix} X_2 & -I \\ I & Z_3^* \end{pmatrix}\right]\text{Pf}\left[\begin{pmatrix} Z_2 & -I \\ I & X_1 \end{pmatrix}\right]\\
&\quad \times \text{Pf}\left[\begin{pmatrix} ((Z_3^*)^{-1} + X_2)^{-1} & -A_2 \\ A_2 & (X_1^{-1} + Z_2)^{-1} \end{pmatrix}\right] \,. \tag{28}
\end{aligned}$$

## B  Diagonal ensemble

The diagonal ensemble is defined as follows. Our initial state is given by $|\psi\rangle = \sum_n c_n|n\rangle$ where the $|n\rangle$ form an orthonormal set of eigenstates. Strictly speaking, the time evolution of our state is then $|\psi(t)\rangle = \sum_n c_n e^{-iE_n t}|n\rangle$. The diagonal ensemble is a density matrix composed of the time-independent diagonal of the initial state density matrix,

$$\rho_D = \sum_n |c_n|^2 |n\rangle\langle n|. \tag{29}$$

In our case, the eigenstates of our system have the form $|\{u\}\rangle \otimes |\{f\}\rangle$ where $|\{f\}\rangle$ are Fock states composed of the single-particle wavefunctions diagonalizing the matter BdG Hamiltonian. Since the flux sectors are orthogonal we can construct a diagonal ensemble within each flux sector. The trace carries over to the extended Hilbert space provided we use the physical subspace projector and our initial state is completely embedded in the physical subspace.

Any operator that changes the flux sector, such as an isolated $\sigma_j^z$, must have a zero expectation value in the diagonal ensemble. The only (possibly) nonzero expectation values of two-spin operators are of of the form $\sigma_{jA}^z \sigma_{jB}^z$ along a $z$-bond. Using $\sigma_A^z \sigma_B^z = i b_A^z c_A i b_B^z c_B = -i u^z c_A c_B$, and the particle-hole transformation defined above, I find

$$\mathrm{Tr}\,\sigma_{jA}^z \sigma_{jB}^z \rho_D = 1 - 2\mathrm{Tr}\,a_j^\dagger a_j \rho_D \tag{30}$$

$$= 1 - 2\frac{1}{N_c} \sum_{\{u\}} \sum_{m=1}^{L} \left( Q_{jm} Q_{mj}^\dagger (R^\intercal R)_{mm} + R_{jm} R_{mj}^\dagger (Q^\intercal Q)_{mm} \right) \tag{31}$$

where I used the diagonalization of the matter Hamiltonian defined in the previous section.

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
