# Peer review of "Quenching the Kitaev honeycomb model"

_SciPost Physics, doi:SciPost Phys. 7, 071 (2019)_

## Round 2 · Referee Report · Anonymous (Referee 1) · 2018-6-6

Strengths

1 . A potentially interesting way of simulating prethermal physics.

Weaknesses

1. How the Neel state is supported on the system eigenstructure has not been clearly worked out.
2. The main claim of a prethermal region in the Toric code limit appears to be a trivial consequence of the Neel state projecting onto the highest energy flat band.
3. It has not been properly argued that thermalization can occur in this model. I am therefore concerned that the notion of prethermaliztion is inaccurate in this instance.

Report

In the paper “Quenching the Kitaev honeycomb model” the author examines how the Neel state time-evolves under various parameters of the Kitaev Honeycomb model. The main claims are that in the gapped Toric-Code phase the Neel state displays pre-thermal behavior, whereas in the gapless B-phase the state quickly relaxes to a valence bond solid.

I have a number of serious concerns/doubts regarding the results and their interpretation.

(1) It has not been proven sufficiently that the Neel state is a superposition of states from all vortex sectors. (see comment A below).
(2) It appears that what the author observes as prethermalization (in the Toric-Code region) is simply the result of the Neel state almost projecting onto a nearly flat energy band. (see comment B below)
(3) Irrespective of the two previous critiques, I am also somewhat concerned about calling the main effect observed as prethermalization.

As a result of these concerns I don’t believe the paper can be published at this time. However the paper is not without merit, and I agree it would be interesting to see if one can use the gauge structure of the model to ‘simulate’ thermalization. However it looks to be like the effect observed here can be explained simply by the flat nature of the spectrum in the Toric code (anisotropic) limit and the fact that, in that limit, the Neel state has a large overlap with the eigenstates in the highest energy band. The observed slow dynamics (in that limit) is a direct consequence of the initial state having a large overlap with eigenstates that have very similar energies.

----Additional comments--------
(A) The author makes the argument that the Neel state overlaps with all vortex sectors. The argument stems from a calculation of the expectation value of the Neel state with an arbitrary plaquette operator. However, although the 0-valued expectation value is a necessary condition, it is not sufficient.
An example of another state that gives a 0-valued plaquette expectation value everywhere is an equally weighted superposition of a vortex-free and a vortex-lattice state. Although I suspect the author is probably correct, is there a way to see that the Neel state is not in a more trivial superposition?

(B) This concern relates to how the Neel state is distributed energetically within each gauge sector.

In the Toric-code limit the ground state manifold is a ferromagnetic arrangement of spins on the z-bonds. Fermionic/matter excitations can be understood as approximately anti-ferromagnetic arrangements of these z-bonds, with an energy cost of around $2 Jz$. Indeed, in the limit that Jx=Jy=0, the Neel state projects exactly onto the fully-filled fermionic state (e.g. the highest energy) in each sector.

As the subsequent dynamical evolution depends on the distribution of supporting eigen-energies , it is therefore natural to expect slow dynamics in the TC limit, especially since, in this limit, the initial state more accurately targets the highest energy flat band. All of the results observed seem to fit with this picture and so I am a bit confused about the main point the author is trying to make.

I should note also that in terms of complexity and eigenstate support, the Neel state is not any more complicated than the its lower-energy ferromagnetic counter-part. Indeed I would expect very similar (perhaps even identical) behavior if this state was chosen instead.

(C) Within each sector, the system is fully described by free-fermions hopping on a lattice. Without a term that breaks the extensive plaquette symmetries, all dynamics will therefore take place within a constrained space within each sector. Doesn’t the free-fermion nature of subsequent dynamics preclude any thermalization?

Additional comments:
(D) Figure 1. The reddish lines seem to disappear behind the legend. They don’t reach the x- or y-axis. Also, it would be interesting to see the results as J_xy approach zero.

(E) The sentence after equation 1 is a subjective one and should be rephrased. Indeed, with respect to Ref. 7, there is much more to that paper than the fermionization method.

(F) On the topic of fermionization methods there are quite a few alternative procedures that make the link between the fermions and spins much more transparent. Kitaev’s methodolgy is excellent if all one requires is the single particle excitation spectrum. However it is cumbersome if one needs to connect back to the actual spin-structure of the eigenstates.

(G) Regarding the concluding comment about topological entanglement entropy arising dynamically. The non-trivial topological entanglement entropy arises exclusively from the gauge degrees of freedom (see Ref 27 and Bray-Ali et al Phys. Rev. B 80, 180504(R) (2009)). Therefore, since all the dynamcis takes place here within each matter sector, there is no way in this set-up for non-trivial entanglement entropy to arise dynamically.

Requested changes

1. The author should include more complete proof for why the Neel state is distributed between all gauge sectors.

2. The author needs to justify why the observed slow dynamics is not simply a consequence of initial state projecting onto a set of eigenstates with similar energy.

3. I would like to see a deeper discussion regarding the notion of how prethermalization can even be valid in this model. Perhaps in the end this concern may just be about terminology. However I have not seen the term “prethermal” used in the context of exactly solvable models before. Can the author provide some references to where this type of scenario has been examined previously? If instead this is a new variation of the idea I think author needs to argue it more forecefully.

4. Figure 1 needs to be fixed (see report)

---

## Round 2 · Referee Report · Anonymous (Referee 2) · 2018-6-28

Strengths

1- The setup of beginning with a product state, quenching and time evolving under a Hamiltonian that hosts topological phases, and looking for signatures of topological order in the final state is an excellent one.

2- Employing the Kitaev honeycomb model and its exactly solvable nature makes for a good approach. The methods used are very nice; expressing the product state in terms of variables used to diagonalize the Kitaev model to solve the quench problem is an innovative addition to the existing literature.

3- The arguments given on the behavior of the nonequilibrium energy content as a function of time as well as those of the short versus long range correlations make for nice physical scenarios.

4 – The treatment of the gauge/plaquette degrees of freedom is well executed and exploits an interesting, key feature of the Kitaev honeycomb.

Weaknesses

1- Overall, the paper can be much more strengthened in terms of the actual physics focus and a clear message being given.

2- With regards to the above, the paper does not fully accomplish what it sets about to do. My biggest issue is with the claim of explicitly addressing topological aspects of quenches in topological systems. Entanglement seems to be one prominent measure evoked here. It is neither the case that entanglement need be the prime measure of topological order (say compared to Chern numbers in non-interacting topological insulators/superconductors) nor need highly entangled states be topological. The Kitaev model does have other measures which are strong indicators.
On that note, there is mention of anyons in the conclusion, but just to say that the highly excited state cannot distinguish signatures. If this were a focus with regards to topology (it is a good one), relevant degrees of freedom (e-m excitations, for instance) should have been defined and tracked more explicitly.

3 – It is difficult to distinguish how much is specific to the Kitaev honeycomb and its topological aspects ,as opposed to something as simple as the transverse Ising chain. See further comments in the Report.

4- See Report for other points on the amibiguity with phase transition; language; referencing; and other issues.

Report

Report
1- Overall, the idea of taking a product state, performing a quench using a Kitaev honeycomb Hamiltonian, and investigating features of the resultant state is very interesting. The question of whether topological aspects evolve is an excellent one ot focus on in this instance.

2- Please see Strength and Weakness sections for further comments on the above and the overall manuscript.

To comment on the manuscript in order of appearance:
3- The Abstract is confusing and similarly, throughout, while there is no problem with grammar, the chain of ideas gets a bit confusing here and there. In the abstract, the computational method is not so important and is confusing. I presume anisoptropy versus not refers to the coupling in the Kitaev honeycomb model, but this is not stated explicitly. The last sentence is not very comprehensible; what signatures of topology are being looked for? What is high energy about the initial state and what implication does this have on topology? There are a nice set of concrete results that could have made its way into the abstract.

4- What does ‘non-topological Hamiltonian’ mean? A Hamiltonian that does not realize a topological phase (ground state property?) for any value of coupling parameters? Perhaps better to refer to a state being topological?

5- It would be insightful to see what aspects of this problem are unique to the Kitaev honeycomb, both in terms of topological aspects as well as its unique gauge and plaquette features. In particular, since entanglement is repeatedly mentioned, how much of what is studied would not be seen in quenches, for instance, in the well explored case of the transverse XY system? Here too, a product state could be used (ground state for just one point in the phase diagram) and then acted upon for its time-evolution by the XY spin Hamiltonian having some specific couplings. Undoubtedly one would obtain a highly entangled, excited state. But likely, many features of spin correlations would be different.
Also, entanglement features are basis dependent. For instance, in the XY case, one could go into a fermionic basis (representing the 1D analog of the 2D p-wave topological superconductor represented by the Kitaev honeycomb model) – can the author comment on this? Particularly given that he uses the lovely transformation initially used by Kitaev.

6- End of Sec. 2 – This discussion on the Neel state, and the gauge-matter representation is very nice and it shows how the power of the Kitaev honeycomb can be used for quenches.

7- Sec. 3 – I am confused about the phase transition aspect. Should one be thinking of crossing a phase boundary in the static Kitaev honeycomb phase diagram? Or more in terms of a Floquet-type dynamic evolution of a phase? A more precise description and measure would be welcome.

8- The thermalization paragraphs on p.5 and the physical arguments are very nice. These could be expanded upon, including starting with standard quench measures (again, say comparing with Ising/XY chains) and how this is unique.

9- Fig. 5 Caption – what is meant by the indication of a particular (valence solid) phase in the long time? Generally one would look for ground state properties. But in this case, due to the quench, the system would be surfeit with excitations.

10 - Could the author make the connection to Drude peaks more precise? In particular, I would imagine one would need to define the conductivity?

11- Please see comments in the Weakness section on anyons.
By looking at the right measures, given the exactly solvable nature of the Kitaev honeycomb, there could perhaps be a very nice non-equilibrium calculation to be done here targetting anyon behavior and strengthening the paper’s premise on quench and topology.

12 - Appendix: Very nice to see a solid, explicit derivation!
After Eq. 12 – How accurate a method is it to average over random gauge fields?

13 - References: There are other highly related works. For instance, even a quick search on the arxiv (say abstract- kitaev honeycomb; abstract – quench).

Requested changes

1- The Weakness and Report sections bring up my main issue with topology, entanglement, etc. This is a major concept that I believe should be addressed throughout the paper.

Beyond that, the Report delineates the various parts where changes would be welcome. To summarize:

2- Abstract

3- Clarify role of entanglement; topological Hamiltonian

4 – Compare with non-topological case, say spin chain; Basis dependence?

5- Clarify notion of phase transition

6- Expand on thermalization discussion

7. Caption of Fig. 5

8.- Discussion of Drude peak

9. Include key discussion on anyons or any other topological aspect.

---

## Round 3 · Referee Report · Anonymous (Referee 1) · 2019-7-27

Strengths

Technically difficult numerical simulations of the Kitaev honeycomb model.

Numerical results that are consistent with what would expect and so I believe the methodolgy is sound and may inspire other studies.

Weaknesses

The author is making a claim of exponentially long time scales which are not backed up by numerical evidence or by what is predicted by the underlying effective models.

Report

The author has made a real effort to expand and explain some of the points I was unsure about. In particular I take the point regarding the growth of entanglement and can see that I was in error here. I also think the numerical method the author is using is giving sensible results and so commend the paper on this technical aspect.

However the paper still suffers from problems. The most serious being the claim that there is some exponentially long time scale with respect to thermalisation.
* * *
In response to the reply:

“The second question is whether these results are not very trivial since we project onto, quote, “a nearly flat energy band”. The Kitaev model is exactly solvable, yes, but it is not a model of free fermions and to speak of “flat energy bands” is not correct. I guess the referee means that the distribution of energy eigenvalues of the initial Neel state is quite narrow, and indeed, this width scales as Var(E)∼J2xy. However, that still begs the question why the expected slower dynamics are not scaling as some power of Jxy? To reiterate, such an exponential dependence on the model parameters is almost a textbook definition of prethermalization.”

I point out that in the paper the author actully says the observed dynamics can be explained by the Toric code, …

“This prethermal regime is governed by an effective high-temperature toric code”

…so I’m not suite sure what to make of some of the sentences above.

I actually agree with the statement that the Toric Code (TC) Hamiltonian governs the so-called "prethermal" regime, which was what I was attempting to pointing out my previous comments.

Lets examine what should happen. In the TC limit (A phases) Kitaev showed that this band of states it is split on the fourth order into $ -C \sum Q_p$ where $C=J_x^2 J_y^2/16 |J_z|^3$ and the Q’s are plaquettes formed out of effective spins on the z-bonds. With respect to the mass gap which is ~ $2J_z$, I would say this set of states is flat, certainly for the system sizes used here, and especially when we are working in the limit of small $J_x$ and $J_y$.

Irrespective of our differing notions of what flat actually means, if the Neel states projected entirely onto the TC band (which with the sign convention used here is actually the lowest energy band in the model) then the dynamics is governed by the relative splittings $4C$ and we would be guaranteed exact recurrences after the relative dynamical phases of all states completed an integer number of cycles e.g. at $T^*=2 \pi /4 C$. The next non-trivial order is the 6th at which case the relative positions of the gauge configurations play some small role (see Phys. Rev. B 78, 245121 (2008)). These and higher order terms probably prevents exact recurrences (within the TC manifold) from occurring.

The T* recurrences are clearly visible in magnetisation plot of Figure 4, with first occurring at pi/2 x 10^4. This makes it clear also that the so-called exponentially long plateaus are simply a consequence of plotting a sinusoidal (e.g cos (t/F)) (for some large F) on a log scale.

Away from the anisotropic limit the dynamics of the matter sector would play a more prominent role, and I would argue that the sudden initial changes in overlap amplitude are due to the much more rapid phase mixing that occurs to the part of the state that is not projected onto the anti-ferromagnetic band.
* * *
With respect to how the Neel state is supported by different gauge sectors the argument in the comments is convincing. Indeed it looks like exactly half of the gauge sectors support the Neel state, where for any of the allowed vortex configurations (those consistent with anti-ferromagnetic configurations) only two of four possible topological sectors are supported.

One important point that occurs here is that (on a torus) there are only 2^(N -3) independent Z_2 gauge configurations in a given topological sector (choice of boundary condition) with the -3 coming from the two topological degrees of freedom and the last because \prod w_p=I. ) While I understand that potentially one can use all u_ij to encode the vortex sector, many of these encodings lead to the same physical scenario. It seems to me that the N_c \approx 2^3N number here seems to be a fairly significant over-counting.

Is it a problem that you haven’t fixed to one gauge convention?
* * *
There is a typo in the definition of the BdG Hamiltonian in the appendix. The way it is written is not Hermitian.

Requested changes

The paper still requires a significant rewrite. In particular the paper cannot suggest that the the prethermal regime is governed by the Toric code when in fact the Toric code predicts something else entirely.

---

## Round 3 · Referee Report · Anonymous (Referee 2) · 2019-8-8

Strengths

1- The problem of quenching and time evolving under a Hamiltonian known to host topological and complex spin phases and obtaining concrete results on such non-equilibrium dynamics is an excellent one to look at. 2- Employing the Kitaev honeycomb model and its exactly solvable nature makes for a good approach. The methods used are very nice; expressing the product state in terms of variables used to diagonalize the Kitaev model to solve the quench problem is an innovative addition to the existing literature.

3- The care taken to address issues, such as doing a more careful study and explaining then notion of the dynamic phase transition, is apparent in the resubmission. A clearer message has been given on quench results between the Neel and valence bond solid phases.

Weaknesses

  1. The main draws of focus on topological and entanglement properties in the quenching procedure have now been removed. But given that they were nebulously stated anyway and now the author does a much more dedicated study of aspects that can be rigorously derived, this weakness is compensated for.

Report

It is a pity that in this resubmission the aspects that I was most excited by – effects of quench on topological and entanglement properties – have now been removed. However, the author has put significant efforts into improving the aspects where he had concrete statements, namely, the behavior of dynamic quantities such as non-equilibrium free energy density and magnetization. The results are non-trivial with regards to non-equilibrium features, though sans dynamic phase transition, and pre-thermalization properties.

There are still some outstanding issues that are to be addressed:

  1. The other referee has brought up excellent points on the statements made on the toric code and pre-thermalization. This would require an involved discussion.

  2. There has been considerable discussion in the reports and response with regards to free fermions, etc. I believe that if the initial state chosen were the zero-flux sector, one could get away with an effectively free fermion treatment (since the Kitaev Hamiltonian conserves flux)? But now, the novelty here is the initial Neel state, which has superpositions of different flux sectors, making the gauge coupling important. The paper would benefit greatly from having a discussion on this point in connection to quenches, given the subtle discussions. Specifically, would there be a scenario, given the right choice of initial state, which would be associated with free fermion evolution? If so, what would the expectation be? How exactly does this deviate from the free fermion problem (there is some nice discussion on what the gauge field is doing but it is buried in the text)? And what striking difference does this give rise to in the quench behavior?

  3. The other referee brought up an interesting comment on Kitaev’s treatment and relating fermions and spin. I too am a fan of Kitaev’s paper as well as the Majorana basis. I am not sure that the referee comment was addressed though, namely, expressing things in the spin basis is a non-trivial task. I think the specific choice of operators, i.e. S^z correlators, and perhaps the magnetization too, overcomes this problem in getting rid of the string in the Majorana basis? Could the author comment on this in the manuscript?

Requested changes

See Report – Please address all three points

---

## Round 3 · Author Response

\section{Referee 1}

I will answer point by point the question from the referee. However, since some points have clear overlap (for example (1), (A) and 1.), I’ve grouped them together to make the discussion as streamlined as possible.

\subsection{Questions: (1) / (A) / 1.}

It is easy to see that $\langle \omega_p \omega_{p’} \rangle = 0$ for $p,p’$ two different plaquettes, because the plaquette operator $\omega_p$ flips the spins on the 1, 2, 4 and 5 sites of the plaquette $p$. To get a non-zero expectation value you need to ‘repair’ such a flip and that cannot be done with a finite number of plaquette operators. Now $\langle \omega_p \omega_{p’} \rangle = 0$ implies that the ‘equally weighted superposition of a vortex-free and a vortex-lattice state’ is not the correct way to describe the Neel state. There is one minor issue, that is actually addressed at the bottom of Page 3: for a finite size system with periodic boundary conditions not all flux configurations contribute. On page 3 I emphasize that the Neel state is an eigenstate of $\sigma^z$ operators. Each such $\sigma^z$ operator changes the gauge/matter configuration, and by systematically applying $\sigma^z$ operators one can probe all the relevant gauge/matter configurations that compose the Neel state. Notice that this leads, in the end, to a superposition of $N_c = 2^{3L_x L_y - L_y}$ different matter-gauge field configurations, and thus to almost all flux configurations. I have extended the discussion on the Neel flux configurations in the final manuscript.

\subsection{Questions: (2) / (3) / (B) / 2. / 3. }

There are two main questions posed here.

  1. The first is regarding the meaning of the phrase ‘prethermalization’. My wording here is completely based on References 15-20, and in particular Ref. 18: "A Rigorous Theory of Many-Body Prethermalization" by Abanin et al. The key ‘strangeness’ of prethermalization is that timescales scale exponentially in the model anisotropy.

Naively, the speed at which some excitation relaxes is some power of the relevant Hamiltonian parameter. The simplest example is just a single-parameter Hamiltonian like the Heisenberg model: double $J$, and the speed of relaxation is halved. Another simple example is the dynamics close to a quantum phase transition, which scales with some (anomalous) power of the separation from the critical point. To have any kind of observable or measurable quantity change exponentially when I change a Hamiltonian parameter linearly is quite unique.

So when does this happen? Well, if you are close to an integrable system, but not just to any integrable system. This is mentioned explicitly on Page 5: the integrable Hamiltonian $H_0$ itself should be a projector model, that is the sum of locally commuting terms. Deviations from such form, so $H = H_0 + \lambda H’$ lead to an exponential time-scale $\mathcal{O}(\lambda)$ in which the integrable part $H_0$ is approximately conserved. I think it’s pretty clear that the Kitaev model in the anisotropic limit satisfies this set-up. It’s a bit strange that the referee asks for “some references” even though I explicitly mention on page 5: "We thus find the emergence, for the anisotropic model, of a distinct prethermalized regime. This can be understood using the framework of Ref. [18].” (Below, at point C, I will also discuss the commonplace confusion about ‘integrability’ and naive preclusion of thermalization itself.)

  1. The second question is whether these results are not very trivial since we project onto, quote, “a nearly flat energy band”. The Kitaev model is exactly solvable, yes, but it is not a model of free fermions and to speak of “flat energy bands” is not correct. I guess the referee means that the distribution of energy eigenvalues of the initial Neel state is quite narrow, and indeed, this width scales as $Var(E) \sim J_{xy}^2$. However, that still begs the question why the expected slower dynamics are not scaling as some power of $J_{xy}$? To reiterate, such an exponential dependence on the model parameters is almost a textbook definition of prethermalization.

Small comment: I agree that choosing an initial ferromagnetic state will lead to the same behavior. I can add this as a comment to the paper.

\subsection{Question: (C) }

On the one hand, the Kitaev model is not a free-fermion model, and it is the interference between different gauge sectors that makes up the time dependence of physical observables. This is made explicit several times throughout the paper, for example in Eqn. (4) where the magnetization (aka a physical observable) is the result of looking at the overlap of time-evolved states in different gauge sectors.

On the other hand, it is a common but very wrong assumption that free systems do not thermalize. Every reasonable translationally invariant system eventually thermalizes to its appropriate generalized Gibbs ensemble or to the relevant diagonal ensemble. Now in the case of the Kitaev model, this means that the final state can be written as a GGE with additional generalized inverse temperatures for the localized plaquette operators - since they are the relevant integrals of motion. The thermalization towards the Diagonal ensemble, see appendix B, is shown in the paper.

\subsection{Question (D) } They don’t ‘disappear’, they are just not included. In Fig. 1 we focus on the long-time behavior, so how the system goes initially from $m=1$ to $m=0.6$ or so is not really the point of the paper. Notice that the data for $J_{xy} = 0.2 J_z$, that is for a higher anisotropy, is included with a finite size scaling in Fig. 5.

On a related note: I noticed an error in the paper, I forgot to include the fact that I used the algorithms of M. Wimmer (\verb+https://arxiv.org/abs/1102.3440+) to compute the pfaffians in Mathematica. I've included this information in the appendix.

\subsection{Question (E) } I agree, basically the whole paper is a gem. I’m happy to rephrase “Kitaev’s genius was his realization” into “Kitaev's key insight was” but in general I’d like to avoid dry writing.

\subsection{Question (F) } I know, I mentioned the brick-wall representation explicitly (Ref. 14). However, for the current purposes I found the method of Kitaev’s majoranas more amenable.

\subsection{Question (G)} I respectfully disagree with the Referee here. Yao and Qi showed that if a state can be written as “gauge-fermion product states” $|u \rangle \otimes | \phi(u) \rangle$, where $|u \rangle$ is the gauge configuration and $|\phi(u) \rangle$ a matter fermion configuration, the resulting entanglement entropy can be split between the entropy generated by the gauge configurations and entropy generated from the fermions. Obviously, this is then true for all eigenstates. However, all bets are off once you include superpositions of such ‘gauge-fermion product states’: on the one extreme, the state suggested by the referee (equally weighted superposition of a vortex-free and a vortex-lattice state) clearly still has topological entanglement entropy, whereas the Neel state obviously has no entanglement whatsoever.

Also, the suggested paper from Bray-Ali does not discuss fermion-gauge models so I don’t understand its relevance for the gauge degrees of freedom.

\subsection{Conclusion} In conclusion, I addressed the two major critiques: \begin{itemize} \item As for prethermalization, I follow the definition of Ref 18. It is clear that the anisotropic Kitaev model satisfies the conditions layed out in that paper. This can explain the otherwise unexpected exponential dependence of the timescales on the anistropy parameter. \item As for the initial Neel state, I showed that also expectation values of products of plaquette operators are zero in the Neel state, thus effectively providing almost all the possible flux configurations that one can have. However, as can be seen on page 3, for systems with periodic boundary conditions not all gauge configurations contribute and the new version will reflect this insight. \end{itemize}

\section{Referee 2}

I will address point by point the issues raised in the section ‘Requested changes’:

\subsection{Question 1.}

The Referee is absolutely right in that in the original manuscript there was no clear discussion of the possible topological aspects of the quench I studied. I thought about this for a long time, and I have decided that it is better to present the work without much focus on the topological aspects. After all, there is no clear measure of topology that I have studied, neither in terms of anyons nor in terms of entanglement. I feel that such a study, while interesting, is more something for a follow-up work.

Therefore, I have rewritten large parts of the text (including the introduction and abstract) to focus on the dynamical transition from a magnetic state to a non-magnetic state. The discussion of a possible dynamical phase transition or crossover, the prethermal regime and the final steady state are better captured in terms of generic quench dynamics. A discussion of the topological aspects is reserved for the final Discussion section.

\subsection{Question 2.} I removed the last sentence and the comment about the method from the abstract, and put more emphasis on the explicit results.

\subsection{Question 3 / Question 9} Following the overall change of emphasis as explained in the answer to Q1, I have added a longer discussion on topological aspects to the Discussion section. I hope I thereby also answered Question 9 satisfactory.

\subsection{Question 4.} The Referee asks here about a comparison between the entanglement structure expected in 'simple' quenches such as the XY model and the quench in the Kitaev model. As answered by Question 1, I have decided to drop the emphasis on entanglement as I could not say anything decisive about it. I feel that therefore there is no also need to discuss entanglement in XY models. I did, however, add some notes on timescales in the transverse field Ising model in response to Question 6.

\subsection{Question 5.} To clarify the notion of a dynamical phase transition, I have added a separate subsection 3.1 focusing completely on the question of whether there is a phase transition when quenching the Kitaev model.

\subsection{Question 6.} I have added a discussion, in section 3.3, showing that for quenches in the transverse field Ising model the typical timescale diverges as a power-law, as is expected in most systems. This emphasises the special nature of the exponential long prethermal regime.

\subsection{Question 7.} It has been shown in Ref.~[26] that a valence-bond solid (VBS) has long-range dimer-dimer correlations. Therefore I have computed this quantity to test whether VBS order existed in the steady state. I have changed the text in the new section 3.3 to reflect the special role of the dimer correlations. Note that measuring a correlation function as a test of long-range order is not restricted to zero temperature.

\subsection{Question 8.} The dynamic two-time spin correlation function as a function of frequency is similar to AC conductivity, which for charged systems can be expressed in terms of the dynamic density-density correlation function. I understand, however, how references to a "Drude peak" can be confusing, so I have removed such mentions and made it clearer what the $\omega = 0$ peak of the correlation function means.

---

## Round 3 · List of Changes

This version is a major revision, see the answers to questions from Referees in the "Author Comments" section.

---

## Round 4 · Referee Report · Anonymous (Referee 1) · 2019-10-12

Strengths

Interesting numerical method to model dynamics of the Kitaev honeycomb model.
Results in the anisotropic regime match what one would expect from the Toric code analysis.

Weaknesses

I'm still a little worried that the so called exponential time scale is shorter that the timescale set by the TC analysis (which is non-exponential).

Report

One remaining concern is that the pre-thermal exponential time scale seems shorter than precise non-exponential time-scale worked out via degenerate perturbation theory. However, the author seems satisfied that there is no contradiction here so I am happy to recommend for publication.

---

## Round 4 · Referee Report · Anonymous (Referee 2) · 2019-11-15

Strengths

  1. As reported earlier, employing the Kitaev honeycomb model and its exactly solvable nature makes for a good approach. The methods used are very nice; expressing the product state in terms of variables used to diagonalize the Kitaev model to solve the quench problem is an innovative addition to the existing literature.

  2. This version of the paper addresses all the issues brought up in previous versions. There is very good data presented on post-quench dynamics. There are solid discussions on a phase transition versus crossover, prethermalization, and the persistence of non-trivial dimer correlations.

Weaknesses

  1. Perhaps there could have been an involved discussion on entanglement and anyon dynamics as highlights of the Kitaev honeycomb. But there is enough rich physics already that this is by no means needed. Also, the author has now added a lovely, thought-provoking discussion on these two topics in the concluding sections.

Report

After the revisions, the author has done a very good job overall of addressing the issues raised by the reviewers. The paper is now a well-presented solid piece of work contributing to the less-studied, important topic of quenches in topological systems. I recommend publication with no reservations.

---

## Round 4 · Author Response

In the list of changes I will comment on each of the referee's excellent comments. I feel that with these changes, the manuscript is finally in publishable form.

---

## Round 4 · List of Changes

Ref 1, main point; and Ref 2-1: Discussion on prethermalization Response: I think I finally find the origin of our misunderstanding, leading hopefully to convergence on our understanding of my results. The ‘prethermal’ regime has two phenomenon that need to be explained: 1) Why does it take exponentially long in Jxy/Jz for the system to relax? 2) Why are there persistent magnetization oscillations in this regime? In the renewed manuscript, I made this distinction clearer. The answer to the first question lays within the realm of prethermalization theory, in particular Ref. [23]. In the new manuscript, I elaborate more on the answer to the second question, which is largely inspired by the referee’s comments. Indeed, starting with an initial Neel state in the toric code, one would expect T=8Pi Jz^3/Jxy^4 periodic oscillations of the staggered magnetization. The essence of the prethermal regime is that the dynamics are * as if * the system is undergoing evolution with a toric code. Indeed, in the new Fig. 6, you can see that this is the case. When J_xy < 0.2 Jz, the oscillations follow almost perfectly the expectations from a toric code. I hope that with this extension, the section 3.2 has become suitable for publication.

Ref 1, second point: "While I understand that potentially one can use all u_ij to encode the vortex sector, many of these encodings lead to the same physical scenario. It seems to me that the N_c \approx 2^3N number here seems to be a fairly significant ..." Response: Actually, it is true that the gauge freedom allows you to construct the Néel state with maybe fewer gauge field configurations than I used. However, the construction I chose has the advantage of being very transparent in that it is a equal-weight superposition of all the possible configurations that satisfy the given criteria. I will add a comment along these lines to the new manuscript.

Ref 2-2: "There has been considerable discussion in the reports and response with regards to free fermions, etc. I believe that if the initial state chosen were the zero-flux sector, one could get away with an effectively free fermion treatment (since the Kitaev Hamiltonian conserves flux)?" This is correct. "But now, the novelty here is the initial Neel state, which has superpositions of different flux sectors, making the gauge coupling important." Again, correct. "The paper would benefit greatly from having a discussion on this point in connection to quenches, given the subtle discussions." I will add a discussion on this at the end of Sec. 2.

Ref 2-3. " I think the specific choice of operators, i.e. S^z correlators, and perhaps the magnetization too, overcomes this problem in getting rid of the string in the Majorana basis? Could the author comment on this in the manuscript?" If I understand the referee correctly, the point is that within my choice of Majorana-Gauge field basis, it is much simpler to compute Sz and correlators of Sz than Sx and Sy-expectation values. I will add a comment on this at the end of Sec. 2.

---

## Editorial Decision

published